# VrdONE: One-stage Video Visual Relation Detection

## ABSTRACT

Video Visual Relation Detection (VidVRD) focuses on understanding how entities interact over time and space in videos, a key step for getting a deeper insight into video scenes beyond basic visual tasks. Traditional methods for VidVRD, challenged by its complexity, usually split the task into two parts: one for identifying what categories are present and another for figuring out their temporal boundaries. This split overlooks the natural connection between these elements. Addressing the need for recognizing entity independence and their interactions across a range of durations, we propose VrdONE, a streamlined yet efficacious one-stage model. VrdONE combines the features of subjects and objects, turning predicate detection into 1D instance segmentation on their combined representations. This setup allows for both category identification and binary mask generation in one go, eliminating the need for extra steps like proposal generation or post-processing. VrdONE facilitates the interaction of features across various frames, adeptly capturing both short-lived and enduring relations. Additionally, we introduce the Subject-Object Synergy (SOS) Module, enhancing how subjects and objects perceive each other before combining. VrdONE achieves state-of-the-art performances on both the VidOR benchmark and ImageNet-VidVRD, showcasing its superior capability in discerning relations across different temporal scales.

## CCS CONCEPTS

• **Computing methodologies → Scene understanding**.

## KEYWORDS

video relation detection, spatiotemporally synergism, set prediction, entity v.s. pair

## 1 INTRODUCTION

Deep learning has propelled significant enhancements in visual video analysis for a variety of tasks such as object tracking [8, 22], action classification [10, 37], and action localization [10, 34, 36]. Despite the advancements, the increasing complexity of video data requires precise interpretation of spatial and temporal relationships among entities in videos. To address this challenge, Video Visual Relation Detection (VidVRD) has been introduced. VidVRD aims to detect all relational instances in a video, each represented by a triplet ⟨*subject, predicate, object*⟩. By harnessing rich semantic insights and interpretability, VidVRD is poised to enhance various

**Unpublished working draft. Not for distribution.**
Permission to make digital or hard copies of all or part of this work for personal or classroom use is granted without fee provided that copies are not made or distributed for profit or commercial advantage and that copies bear this notice and the full citation on the first page. Copyrights for components of this work owned by others than the author(s) must be honored. Abstracting with credit is permitted. To copy otherwise, or republish, to post on servers or to redistribute to lists, requires prior specific permission and/or a fee. Request permissions from permissions@acm.org.
*ACM MM, 2024, Melbourne, Australia*
© 2024 Copyright held by the owner/author(s). Publication rights licensed to ACM.
ACM ISBN 978-x-xxxx-xxxx-x/YY/MM
https://doi.org/10.1145/nnnnnnn.nnnnnnn

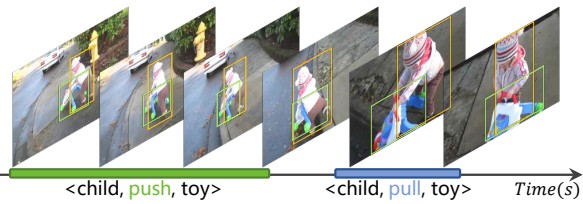

<child, push, toy>    <child, pull, toy>    *Time(s)*

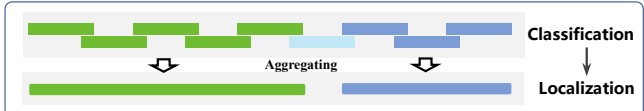

**(a) Clip-level Classification-based Pipeline**

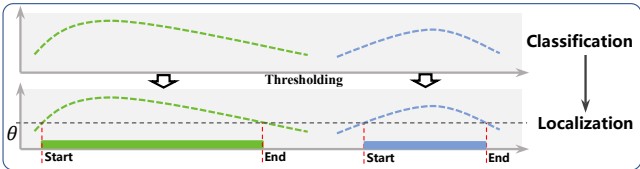

**(b) Frame-level Classification-based Pipeline**

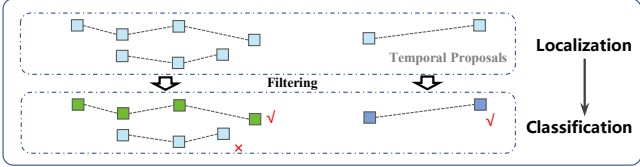

**(c) Localization-based Pipeline**

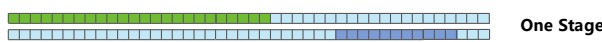

**(d) Our 1D Temporal Instance Segmentation Paradigm**

**Figure 1: Classical pipelines in existing VidVRD methods include: (a) clip-level classification-based, (b) frame-level classification-based, and (c) localization-based approaches. These methods often overlook the spatiotemporal interactions between entities, thus failing to fully capture both transient and persistent relations. In contrast, our approach (d) utilizes a 1D temporal instance segmentation formulation that concurrently facilitates relation classification and frame-level relation mask generation for all relations in a single step, eliminating the need for additional post-processing.**

downstream applications, including video captioning [42], video question answering [42], and video visual grounding [15].

The VidVRD framework is divided into three sub-tasks: entity tracking, relation classification, and temporal boundary localization. As illustrated in Fig. 1, the process begins with the identification of each entity's category and spatial location using pretrained video tracking models [6]. Traditional approaches to VidVRD typically treat the tasks of classification and temporal localization

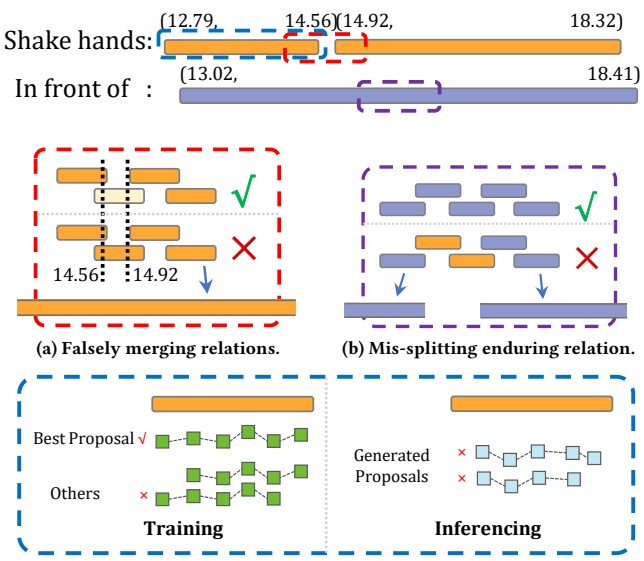

(a) Falsely merging relations.

(b) Mis-splitting enduring relation.

(c) Throwing all imperfectly matchable proposals.

**Figure 2: Limitations of existing two-stage methods. In classification-based methods, heuristic aggregation can lead to incorrect temporal localizations, causing (a) consecutive relations to be mistakenly identified as a single relation, and (b) long-lasting relations to be improperly split into shorter segments. Localization-based methods also have drawbacks, where (c) relations might go undetected during inference due to mismatches with the fixed-length proposals.**

as distinct, processing them sequentially in either a classification-based or localization-based manner. In classification-based strategies [30, 43], relations are first identified on a clip-level (Fig. 1(a)) or frame-level(Fig. 1(b)), and relation periods are determined using heuristic temporal aggregation algorithms [5, 30]. Conversely, localization-based approaches (Fig. 1(c)) start with generating temporal proposals, which are then refined through a redundancy filtering mechanism before classification.

However, existing methods do not coherently account for the spatiotemporal interactions between entities, resulting in suboptimal performance in both relation classification and localization. On one hand, the integration of clip-level and frame-level short-term relations primarily depends on locally extracted features. This can lead to ambiguous detections at the temporal boundaries of relations, such as mistakenly splitting a continuous relation into two disjoint ones (Fig. 2(a)) or improperly merging temporally adjacent relations of the same category (Fig. 2(b)). On the other hand, the use of generated proposals creates fixed-length temporal templates for video relations. As depicted in Fig. 2(c), these templates often overlook potential relations that do not perfectly align with them during the inference stage, thereby constraining their effectiveness.

In real-world scenarios, object interactions exhibit varied patterns across spatial and temporal dimensions. As shown in Fig. 3, each type of video relation in the VidOR dataset [29] displays distinct spatiotemporal characteristics, including differences in duration and frequency. Furthermore, entities within these relations

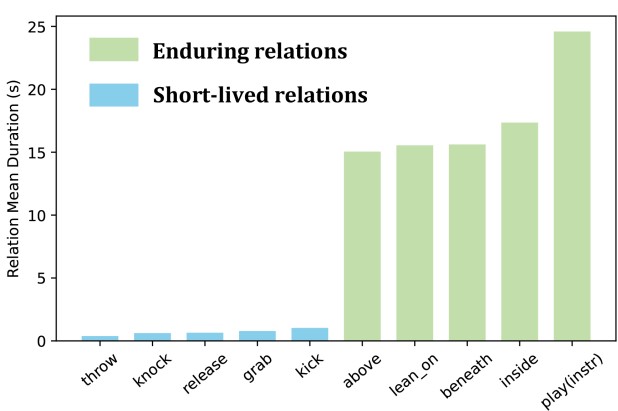

**Figure 3: Distributions of all relations in the VidOR dataset.**

vary in aspects such as movement speed and range. For instance, the relation "in front of" and "shake hands" might occur simultaneously between two individuals during the same video segment. While "in front of" might persist throughout the segment, "shake hands" typically lasts only a few seconds and involves rapid movement. This diversity in spatiotemporal dynamics underscores the importance of accounting for these variations to accurately categorize relation types. Motivated by these observations, we aim to improve our model's performance in video relation detection by integrating richer spatiotemporal information.

Building on this concept, we aim to integrate video relation classification and temporal boundary localization into a single holistic problem, reformulating it as a 1D temporal instance segmentation task (see Fig. 1(c)). This unified approach allows for more precise relation classification and detailed relation boundary localization within a single inferencing step, benefitting from the improved supervision provided by temporal location binary masks.

In this context, we introduce VrdONE, a spatiotemporal synergistic transformer designed for one-stage video visual relation detection. This model efficiently detects all relation instances between subject-object pairs in an untrimmed video. Initially, we capture the temporal and spatial features of all entities in the video sequence. For each subject-object pair, we align their features along the temporal dimension to enhance spatiotemporal interactions across various frames. We have developed the Subject-Object Synergy (SOS) module to improve mutual perception between the subjects and objects. Additionally, a Bilateral Spatiotemporal Aggregation (BSA) mechanism has been designed to effectively learn features that encapsulate both transient and persistent relations. These features are then processed by a relation encoder and directed towards the classification and temporal boundary localization branches. Both branches are concurrently trained in a single stage, supported by a relation identification loss and a mask prediction loss.

In summary, our contributions are threefold:

- We offer a novel perspective on the Video Visual Relation Detection (VidVRD) challenge by reformulating it as a 1D instance segmentation task. This innovative approach allows

for simultaneous category identification and binary mask generation for video relations in a single processing step.

- We propose VrdONE, a unique one-stage framework for VidVRD. Through the use of Bilateral Spatiotemporal Aggregation, VrdONE enhances the interaction between subjects and objects across time and space, effectively capturing both transient and long-lasting relations.
- Our experimental results on various benchmarks confirm that VrdONE sets a new standard for VidVRD. It significantly improves upon the state-of-the-art in both relation classification and temporal boundary localization.

## 2 RELATED WORK

**Video Visual Relation Detection.** Recent advancements in Video Visual Relation Detection (VidVRD) primarily fall into two categories: classification-based and localization-based methods. Utilizing features from pretrained tracking models [6], Shang *et al.* [31] developed the first classification-based pipeline. This approach segments videos into clips for short-term relation classification and employs an association algorithm for temporal localization. Subsequent studies [14, 30, 31, 38, 41] have refined this method by enhancing classification accuracy using graph convolution networks [24, 38] or integrating multi-modal features [32, 38]. Innovations in association algorithms by Wei *et al.* and Su *et al.* [32, 38] have led to more precise temporal localization. However, clip-based approaches struggle with prolonged relations and are prone to errors from cumulative association steps. To better capture long-range relations, Chen *et al.* [5] introduced a multi-modal prototype learning approach that uses a 1D watershed algorithm [27] for frame-level classification and temporal localization. Concurrently, Gao *et al.* [11] and Zheng *et al.* [43] have explored parallel learning strategies for spatial and temporal relation metrics. Contrarily, Liu *et al.* [18] have attempted a new direction by generating numerous temporal proposals through sliding windows, filtered by template matching to pinpoint relation durations.

Differing from these approaches, we reconceptualize the challenges of classification and temporal localization into a unified 1D instance segmentation task within a one-stage framework. Our method leverages interactions between subject and object features across frames to effectively capture both transient and persistent relations, significantly improving the precision of relation classification and localization.

**Spatiotemporal Synergistic Learning in Videos.** Understanding vision tasks in videos requires a spatiotemporal synergistic approach. Initially, 3D convolutional neural networks were used to extract features across both spatial and temporal dimensions [4, 10]. More recently, transformer architectures have brought significant advancements in computer vision [9, 19, 35]. For instance, ViViT [1] integrates these architectures into video processing and sets new performance benchmarks, surpassing older 3D convolution-based methods. The Video Swin Transformer [20] adapts the Swin Transformer concept to video by expanding it into three dimensions, which enhances information capture from local to global contexts, improving efficiency in learning. Similarly, VideoMAE [34] and its successor, VideoMAE V2 [36], leverage a Masked AutoEncoder approach in a self-supervised learning framework, applying consistent spatial masks across video clips to increase model robustness and effectiveness, thereby achieving notable performance improvements in various video processing tasks. Overall, integrating spatiotemporal elements is crucial for optimizing video processing across diverse applications.

## 3 METHOD

### 3.1 Preliminaries

**Problem Setting.** Given an untrimmed video $V$ of length $L$, which contains $N$ entities and $M$ possible relations, the goal of VidVRD is to learn a video relation detector $\mathcal{G}$ to generate all possible relations between the entities in $V$ and their corresponding durations, such that

$$\mathcal{G}(F) = \{(\langle S_i, R_k, O_j \rangle, T_{start}, T_{end})\}, i, j \in [1, N], k \in [1, K], \quad (1)$$

where $S_i$ and $O_j$ denotes the subject and object when relation $R_k$ happens, $T_{start}$ and $T_{end}$ denotes the begin and the end of $R_k$. In this case, $F = \{f_1, f_2, ..., f_N\}$ represents the extracted features of the tracklets for all the objects. The feature for object $i$ is represented by $f_i \in \mathbb{R}^{l_i \times C}$, which is a set of feature vectors extracted for uniformly-sampled consecutive video frames, where $l_i \leq L$ is the period that object exists in $V$. Typically, the pipeline of VidVRD is divided into three sub-tasks: entity tracking, relation classification, and temporal boundary localization. After extracting $F$ from the results of entity tracking, previous works often treat the relation classification and temporal boundary localization of predicates separately. This procedure can be explained by Bayes's Formula, such that the distribution of the relation type $R_c$ and its duration $R_d$ are formulated either as:

$$P(R_c, R_d|F) = P(R_d|R_c, F)P(R_c|F), \quad (2)$$

or

$$P(R_c, R_d|F) = P(R_c|R_d, F)P(R_d|F). \quad (3)$$

However, the ignorance of the inherent connection between the two tasks and consequently deteriorates both the classification and localization performance. To fully mitigate the spatial and temporal features during the interaction of subjects and objects, we propose to reformulate the problem in a one-stage manner, *i.e.*, directly estimating $P(R_c, R_d|F)$.

**Attention Mechanism in Transformers** [35] has demonstrated its great ability to capture global information along an input sequence. Given the input query, key, and value, denoted by $q \in \mathbb{R}^{l_q \times D_q}, k \in \mathbb{R}^{l_k \times D_k}, v \in \mathbb{R}^{l_v \times D_v}$, the attention operation is calculated as:

$$\text{Attn}(q, k, v) = \text{Softmax}(\frac{q \cdot k^T}{\sqrt{D_q}}) \cdot v, \quad (4)$$

where typically $D_q = D_k, l_k = l_v$. Among the vanilla attention architecture, the self-attention proposes to generate the $q, k, v$ from the same input sequence $e \in \mathbb{R}^{l_e \times D}$ with three projection function:

$$\sigma_q(e) = e \cdot W_q, \ \sigma_k(e) = e \cdot W_k, \ \sigma_v(e) = e \cdot W_v. \quad (5)$$

where $W_q \in \mathbb{R}^{D \times D_q}$, $W_k \in \mathbb{R}^{D \times D_k}$, and $W_v \in \mathbb{R}^{D \times D_v}$ are the coefficients of the three projection functions.

**Local Attention.** To capture local information within the neighboring region of the input sequence, local attention is proposed for

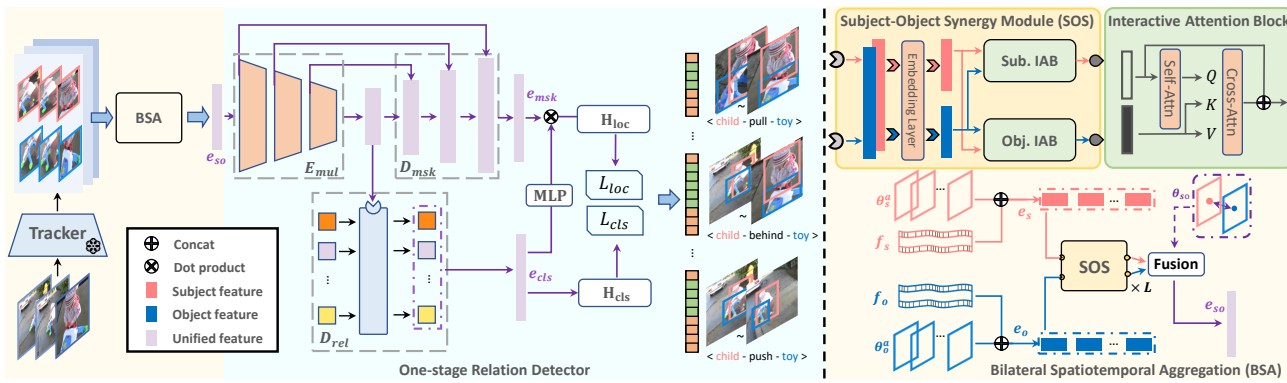

**Figure 4: The pipeline of our VrdONE. Given an untrimmed video, we obtain the temporal and spatial feature ($f$ and $\theta$) for all of entities' traklets using a frozen pretrained video tracker. For each subject-object pair, we apply the Bilateral Spatiotemporal Aggregation (BSA) to encapsulate both information from transient and persistent relations into the feature embeddings, proceeding them through $L$ Subject-Object Synergy (SOS) modules. After equipping the enriched embeddings with the relative spatial movement $\theta_{so}$, the resulted unified embeddings $e_{so}$ is further processed by the relation encoder $E_{mul}$ and directed to two synergistic decoder $D_{msk}$ and $D_{rel}$. With the help of the generated temporal-aware feature $e_{msk}$ and category-aware feature $e_{cls}$, VrdONE finally achieves one-stage precessing for both video relation classification and temporal localization.**

restricting the perceptive fields on the sequence. Before applying the projection function, the input query, key, and value $e_q, e_k, e_v$ will be separately divided into $I$ small segments $e^t, t \in [1, I]$, each segment is processed by an independent 1D convolutional layer $\hat{e}^t = \textbf{Conv1D}(e^t)$. The attention calculation is also performed segment-wise as:

$$\textbf{LocalAttn}(e_q^t, e_k^t, e_v^t) = \textbf{Softmax}(\frac{\sigma_q(\hat{e}_q^t) \cdot \sigma_k(\hat{e}_k^t)^T}{\sqrt{D_q}}) \cdot \sigma_v(\hat{e}_v^t). \quad (6)$$

The results of $I$ segments will be further concatenated into one for the next attention layer. Based on this definition, we define the utilized local self-attention layer and local cross-attention layer used in our VrdONE as

$$\textbf{LocalSA}(e^t) = \textbf{Softmax}(\frac{\sigma_q(\hat{e}^t) \cdot \sigma_k(\hat{e}^t)^T}{\sqrt{D_q}}) \cdot \sigma_v(\hat{e}^t), \quad (7)$$

and

$$\textbf{LocalCA}(e_s^t, e_o^t) = \textbf{Softmax}(\frac{\sigma_q(\hat{e}_s^t) \cdot \sigma_k(\hat{e}_o^t)^T}{\sqrt{D_q}}) \cdot \sigma_v(\hat{e}_o^t), \quad (8)$$

respectively.

### 3.2 Overview

The goal of VrdONE is to build an efficacious one-stage video relation detector for simultaneously handling the relation classification and temporal localization. The overall pipeline of VrdONE is shown in Fig. 4. Firstly, we apply a pretrained object detector [6, 26] to extract objects' features $F$ together with their spatial position $\Theta = \{\theta_1, \theta_2, ..., \theta_N\}$. For each subject-object pair, we process their features $(f_s, \theta_s, f_o, \theta_o)$ using the Bilateral Spatiotemporal Aggregation (Section 3.3) for fully perceiving spatiotemporal interactions in the video. Specifically, we propose a subject-object synergy module for improving the mutual perception between the two entities. The

resulting unified embedding $\theta_{so}$ is further proceeded to the one-stage relation detector (Section 3.4) for both relation classification and temporal boundary localization. The one-stage relation detector consists of a relation encoder $E_{mul}$, a relation decoder, and a temporal mask decoder $D_{msk}$, and concurrently trained in a single stage by a relation identification loss and a mask prediction loss.

### 3.3 Bilateral Spatiotemporal Aggregation

In Bilateral Spatiotemporal Aggregation (BSA), we promote the mutual awareness of the subject and object features through mutual perception and ultimately encode them into a unified relational representation for later time dimensional segmentation.

Given a pair of consecutive and untrimmed subject and object features, we generate subject-object pairs by enumerating all two-by-two combinations of triplet proposals, forming the set $\mathcal{P} = \{(f_s, f_o)_n | 1 \leq n \leq N * (N-1)\}$, where $f_s$, $f_o$, and $N$ denotes the subject feature, the object feature, and the number of detected entities. Subsequently, we crop the subject-object pairs with the overlapping time range to get the synchronized feature vectors $f_s \in \mathbb{R}^{l_{so} \times C}$ and $f_o \in \mathbb{R}^{l_{so} \times C}$, with $l_{so}$ denoting the length.

To embed the detected spatial information into the features, we adopt an approach from [11] and employ absolute positional representations $\theta^a \in \mathbb{R}^{l_i \times 8}$ for each entity. To be specific, the positional representations are comprised of the normalized bounding bbox coordinates and the offsets between two consecutive frames. Thereafter, the visual features $f$ and spatial features $\theta^a$ are integrated into a general entity embedding with a multilayer perceptron (MLP), formulated as:

$$e = \textbf{MLP}(\textbf{Concat}(f, \theta^a)), \quad (9)$$

where $\mathbf{Concat}(\cdot, \cdot)$ represents the concatenation along feature dimensions. Following the above process, the visual and spatial features of both subject and object are integrated into entity embeddings $e_s$ and $e_o$, which are further fed into the Subject-Object Synergy Module to comprehend interactions.

**Subject-Object Synergy Module.** The Subject-Object Synergy (SOS) module facilitates interaction between subject and object features to enhance mutual understanding. The SOS module is composed of an embedding layer and two Interactive Attention Blocks (IAB).

The embedding layer shares the same structure as the encoder layer of a vanilla Transformer, consisting of a local multihead self-attention and an MLP. Specifically, the embedding layer of the $l_{th}$ SOS block is defined as:

$$\bar{e}^l = \mathbf{LocalSA}(e^{l-1}) + e^{l-1},$$
$$\hat{e}^l = \mathbf{MLP}(\bar{e}^l) + \bar{e}^l. \tag{10}$$

Accordingly, the subject and object features are embedded and are denoted as $\bar{e}^l$ and $\hat{e}^l$.

Within the SOS module, the Interactive Attention Block (IAB) enables information exchange between subject and object features to enrich their representations. Concretely, the Interactive Attention Block is composed of a self-attention layer and a cross-attention layer. For instance, to integrate object features into subject features, the aggregated subject representation is expressed as:

$$\tilde{e}_s^l = \mathbf{LocalSA}(\hat{e}_s^l),$$
$$e_s^l = \mathbf{LocalCA}((\tilde{e}_s^l, \hat{e}_o^l)) + \hat{e}_s^l. \tag{11}$$

Likewise, we augment the object features with the mutual information from the subject and finally obtain an aggregated object feature $e_o^l$.

After applying $L$ SOS layers, the enhanced subject and object features ($e_s^L$ and $e_o^L$) capture comprehensive representations with innovative features from the interactions. We then fuse the subject and object features to form a unified representation for the subject-object relation. To further facilitate positional awareness, we inject the relative position $\theta_{so}^r$ as follows:

$$\theta_{so}^r = [s_x, s_y, s_w, s_h, s_a],$$
$$= [\frac{x^s - x^o}{x^o}, \frac{y^s - y^o}{y^o}, \log\frac{w^s}{w^o}, \log\frac{h^s}{h^o}, \log\frac{w^s \cdot h^s}{w^o \cdot h^o}]. \tag{12}$$

Finally, the subject feature, the object feature, and the relative position are projected to form the final representation of the subject-object relation $e_{so}$ using a two-layer MLP as:

$$e_{so} = \mathbf{MLP}(\mathbf{Concat}(e_s^L, e_o^L, \theta_{so}^r)). \tag{13}$$

## 3.4 One-stage Relation Detector

After obtaining the unified embedding $e_{so}$ that contains rich spatiotemporal information, we further process it to achieve one-stage relation classification and temporal localization through the one-stage relation detector. The one-stage relation detector is composed of a Relation Encoder $E_{mul}$, relation decoder $D_{rel}$, and temporal mask decoder $D_{msk}$.

**Relation Encoder.** We follow the design of feature pyramid network [16] and implement our relation encoder to capture multiscale features over varying temporal lengths. The relation encoder is stacked by a series of transformer blocks, which share a similar architecture with blocks defined in Eq. 10. Additionally, we propose to downsample the features before inputting them into each transformer block to perceive more long-range temporal information. By treating the unified features embedding $e_{so}$ as the input of the first encoding block, the calculation of each block can be formulated as:

$$\hat{a}^{l-1} = \delta(a^{l-1}),$$
$$\bar{a}^l = \mathbf{LocalSA}(\hat{a}^{l-1}) + (\hat{a}^{l-1}), \tag{14}$$
$$a^l = \mathbf{MLP}(\bar{a}^l) + \bar{a}^l,$$

where $a^{l-1}$ is the output of the previous block and $\delta$ is the downsampling operation. In this way, multi-scale spatiotemporal features can be obtained from different layers of the relation encoder, forming a feature pyramid $\mathbf{A} = \{a^1, a^2, ..., a^{L_e}\}$, where $L_e$ is the number of transformer blocks.

**Relation Decoder and Temporal Mask Decoder.** We employ a query-based transformer as our relation decoder and a feature pyramid decoder for temporal mask generation.

For the relation classification, our relation decoder receives $a^{L_e}$ as its input to access high-dimensional semantic information. Specifically, the relation decoder consists of $L_{rel}$ transformer blocks with $N_q$ learnable query embeddings $q \in \mathbb{R}^{N_q \times d}$, which serve as template learners for all possible relation instances within a video. $N_q$ and $d$ denote the number and dimension of query embeddings. The calculation can be formulated as:

$$q^l = \mathbf{LocalSA}(q^{l-1}),$$
$$e_{rel}^l = \mathbf{LocalCA}(q^l, e_{rel}^{l-1}), \tag{15}$$

where $e_{rel}^1 = a^{L_e}$. The final output of relation decoder $e_{cls} = e_{rel}^{L_{rel}}$ will pass through a classification head $H_{cls}$ to output the categories of the detected relations.

For temporal relation localization, we generate a fine-grained mask using the temporal mask decoder for precise relation boundary detection in a per-frame mode. The temporal mask decoder contains a series of lateral connection layers for progressively upsampling the pyramid feature $\mathbf{A}$. The number of lateral connection layers is the same as the number of transformer blocks in $E_{mul}$. Concretely, the feature aggregation in the $l$-th layer is

$$\tilde{a}^l = \mathbf{Conv1D}(\eta(\tilde{a}^{l-1}) + \mathbf{Conv1D}(a^l)), \tag{16}$$

where $\eta$ denotes the upsampling operation, which performs linear interpolation on $\tilde{a}^{l-1}$. The decoder's output $e_{msk}$ is recovered to the same length with $e_{so}$ for better perception of temporal variations, and finally incorporates the classification embedding $e_{cls}$ through the localization head $H_{loc}$ to generate per-frame relation mask.

## 3.5 Training and Inference

**Loss Functions.** Similar to MaskFormer [7], we employ a Bipartite Matching strategy to assign different queries to learn the corresponding instances. The matching cost for relation classification

Table 1: Comparison with state-of-the-arts on VidOR dataset. For detectors, FR, MG, and IE symbolize Faster R-CNN [26], MEGA [6], and Integrated Encoder, respectively. For extra features, L and M denote Language and Mask features, whereas I3D [4] and CLIP [25] denote visual feature extractor. For Social Fabric and our VrdONE, we represent the variants with extra features with a "-X" postfix. The best and second best performances are bolded and underlined.

| Method | Detector | Extra Feature | Relation Detection | | | Relation Tagging | | |
|---|---|---|---|---|---|---|---|---|
| | | | mAP | R@50 | R@100 | P@1 | P@5 | P@10 |
| VRD-STGC [18] | FR | – | 6.85 | 8.21 | 9.90 | 48.92 | 36.78 | – |
| IVRD [14] | FR | – | 7.42 | 7.36 | 9.41 | 53.40 | 42.70 | – |
| TSPN [40] | FR | – | 7.61 | 9.33 | 10.71 | 53.14 | 42.22 | 34.94 |
| VIDVRD II [30] | FR | – | 8.65 | 8.59 | 10.69 | 57.40 | 44.54 | 33.30 |
| BIG [11] | MG | I3D+L | 8.54 | 8.03 | 10.04 | 64.42 | 51.80 | 40.96 |
| HCM [38] | MG | – | 10.44 | 9.74 | 11.23 | 67.43 | 52.19 | 40.30 |
| VRDFormer [43] | IE | – | 11.19 | 11.05 | 13.34 | 63.71 | 51.07 | 39.89 |
| Social Fabric [5] | FR | I3D | 9.54 | 8.49 | 10.17 | 59.24 | 47.24 | 35.99 |
| Social Fabric-X [5] | FR | I3D+L+M | 11.21 | 9.99 | 11.94 | **68.86** | 55.16 | 43.40 |
| **VrdONE** | MG | – | 11.86 | 11.13 | 14.21 | 66.11 | 54.92 | 43.90 |
| **VrdONE-X** | MG | CLIP | **12.17** | **11.41** | **14.55** | 67.67 | **55.58** | **44.28** |

and mask prediction is denoted as

$$\mathcal{L}_{matching} = \lambda_{cls} \cdot \text{CE}(\hat{p}_i, c_j^{gt}) + \mathcal{L}_{mask}(\hat{m}_i, m_j^{gt}), \quad (17)$$

where the classification cost $-\hat{p}_i(c_j^{gt})$ used in DETR [3] is replaced by Cross Entropy loss. This substitution is made perhaps due to the fact that $-p_i(c_j^{gt})$ incurs a higher cost than cross entropy, potentially leading to premature overfitting in the training process, thereby hindering our model's learning. The $\mathcal{L}_{mask}$ is

$$\mathcal{L}_{mask} = \lambda_{mf} \cdot \text{FL}(\hat{m}_i, m_j^{gt}) + \lambda_{md} \cdot \text{Dice}(\hat{m}_i, m_j^{gt}), \quad (18)$$

which is a binary focal loss [17] and a dice loss [23] respectively.

The overall loss function for training is given by:

$$\mathcal{L} = \lambda_{cls} \cdot \text{CE}(\hat{p}_{\sigma(i)}, c_i^{gt}) + \mathbb{I}_{c_i^{gt} \neq \varnothing} \mathcal{L}_{mask}(m_{\sigma(i)}, m_i^{gt}), \quad (19)$$

where $\sigma(i)$ denotes the index of the query matched to the ground truth with index $i$.

**Inference Phrase.** During testing, we exhaustively enumerate all possible pairs to detect relations within the current video, resulting in $N \times (N-1)$ potential subject-object pairs for inference. However, our model is capable of parallelly detecting all possible subject-object pairs and outputting all detection results in one step. For segmented frames, we consider those with a foreground probability greater than 0.5 as the detected relation range. Any post-processing to avoid isolated noisy positive points is ignored, as we have observed that our model demonstrates robustness in accurately identifying the temporal boundaries of relation instances.

## 4 EXPERIMENTS

**Datasets.** To evaluate our method, we conduct experiments on two datasets: ImageNet-VidVRD [31] and Video Object Relation (VidOR) [29]. ImageNet-VidVRD comprises 1,000 videos sourced from the ILSVRC2016-VID dataset [28], with a total duration of approximately 3 hours. It contains 35 entity categories and 132 relation categories. Annotations in ImageNet-VidVRD are coarsely labeled

with relation lengths as multiples of 15 frames, while entity tracklets are densely annotated in each frame to form ⟨*subject, predicate, object*⟩ triplets. The dataset is split into 800 training videos and 200 testing videos. The VidOR dataset consists of 10,000 user-generated videos selected from YFCC-100M [33], totaling approximately 98.6 hours. There are 80 entity categories and 50 predicate categories. VidOR is partitioned into a training set with 7,000 videos, a validation set with 835 videos, and a testing set with 2,165 videos. Following standard practice, we train our model on the training set and test on the validation set. Unlike ImageNet-VidVRD, which has sparse annotations, VidOR provides densely labeled relations on the temporal dimension, demanding more precise reasoning capabilities. Additionally, as depicted in Fig. 3, the mean durations of relations in VidOR are typically much longer than those in ImageNet-VidVRD and vary across relation categories, posing additional challenges.

**Evaluation Metrics.** We assess VrdONE's performance on two tasks: (1) Relation Detection (RelDet): This task involves detecting a set of visual relation triplets, and the corresponding tracklets of subject and object. A detected triplet is deemed correct if suffices both matching the ground-truth triplet and the detected tracklets manifest sufficient overlap with the ground-truth, *e.g.*, $vIoU > 0.5$. We utilize mAP and Recall@K (R@K, K=50, 100) as the metrics for RelDet. (2) Relation Tagging (RelTag): This task only solely evaluates the precision of visual relation triplets and disregards the localization results of tracklets. Precision@K (P@K, where K=1, 5, 10) is employed as the evaluation metric for RelTag.

**Implementation Details.** Following [11, 38], we utilize the pre-trained Object Detector MEGA [6] with backbone ResNet-101 [12]. Detection results are consolidated into object tracklets using deepSORT [39]. We set the maximum length of overlapped subject-object durations as 512, otherwise cut out the outer length. The Multi-scale Transformer Encoder incorporates 3 blocks, alongside the output from SOS, resulting in a 4-layer feature pyramid. With a downsampling ratio of 2, the feature pyramid comprises lengths of [512, 256, 128, 64] respectively. The decoder consists of 4 layers,

**Table 2: Comparison with state-of-the-arts on VidVRD dataset. $^{\dagger}$ denotes the version implemented by the authors.**

| Method | Detector | Extra Feature | Relation Detection | | | Relation Tagging | | |
|---|---|---|---|---|---|---|---|---|
| | | | mAP | R@50 | R@100 | P@1 | P@5 | P@10 |
| VRD-STGC [18] | FR | I3D | 18.38 | 11.21 | 13.69 | 60.00 | 43.10 | 32.24 |
| IVRD [14] | FR | – | 22.97 | 12.40 | 14.46 | 68.83 | 49.87 | 35.57 |
| TSPN [40] | FR | – | 18.90 | 11.56 | 14.13 | 60.50 | 43.80 | 33.73 |
| Social Fabric [5] | FR | – | 19.23 | 12.74 | 16.19 | 57.50 | 43.40 | 31.90 |
| Social Fabric-X [5] | FR | I3D+L+M | 20.08 | 13.73 | 16.88 | 62.50 | 49.20 | 38.45 |
| VIDVRD II$^{\dagger}$ [30] | FR | – | 23.85 | 9.74 | 10.86 | 73.00 | 53.20 | 39.75 |
| BIG [11] | MG | – | 26.08 | 14.10 | 16.25 | 73.00 | 55.10 | 40.00 |
| HCM [38] | MG | – | 29.68 | 17.97 | 21.45 | 78.50 | 57.40 | 43.55 |
| **VrdONE** | MG | – | **31.33** | **18.20** | **21.61** | **80.50** | **59.40** | **44.17** |

with the number of queries $N_q$ set as 9. Parameters $\lambda_{cls}, \lambda_{mf}, \lambda_{md}$ are set to 2, 2, and 5.

Prior to Local Attention and MLP computation, calculating Local Attention and MLP, LayerNorm [2] is implemented. Drop-out and Drop-path [13] rates are specified as 0 and 0.1. Training of VrdONE employs the AdamW [21] optimizer with a learning rate of $2 \times 10^{-4}$. Warmup and Exponential Moving Average (EMA) techniques are employed to enhance and stabilize the training process.

## 4.1 Comparison with State-of-the-Arts

We conduct experiments on ImageNet-VidVRD and VidOR datasets and compare our VrdONE with the state-of-the-art methods on RelDet and RelTag tasks, as illustrated in Table 1 and Table 2.

On the VidOR dataset, we implement two versions VrdONE detector with the ordinary one obeys a traditional pipeline while the extra version incorporates features extracted by the CLIP [25] image encoder. For the ordinary implementation, our vanilla VrdONE achieves state-of-the-art performance on four metrics. In particular, VrdONE exhibits a noticeable improvement (+0.67%, +0.08%, and +0.87%) on all the RelDet metrics compared to the previous state-of-the-art [38, 43], indicating a comprehensive enhancement in the temporal boundary localization by leveraging the spatiotemporal interaction. As for the implementation with additional CLIP features, method [11] and implementation [5] with extra features are also involved for a fair comparison. With extra CLIP [25] features integrated, VrdONE achieves the best or second-best performance across all six metrics. Specifically, VrdONE balances tasks between RelDet with RelTag, maintaining robust relation classification performance comparable to specialized models like Social Fabric [5], which excel in perceiving category relationships. Notably, VrdONE demonstrates a significant advantage in temporal boundary localization performance, showing improvements of +0.96%, +0.36%, and +1.21% on mAP, R@50, and R@100, respectively.

On the VidVRD dataset, VrdONE outperforms HCM by +1.17%, +0.23%, +0.16%, +2.00%, +2.00%, and +0.62% in terms of all the RelDet and RelTag metrics. By amalgamating the diverse metrics across both datasets, our VrdONE demonstrates exceptional performance, thereby validating the efficacy of the single-step methodology.

**Table 3: Ablation of Subject-Object Synergy (SOS) module. "w/o SOS" denotes the removal of SOS module. "w/o IAB", Cross" and "IAB" indicate SOS with the removal of IAB module, basic cross-attention, and Interactive Attention Block, respectively. $^{*}$ indicates our implementation.**

| Approach | Relation Detection | | | Relation Tagging | | |
|---|---|---|---|---|---|---|
| | mAP | R@50 | R@100 | P@1 | P@5 | P@10 |
| w/o SOS | 11.28 | 10.83 | 13.64 | 65.74 | 54.68 | **44.06** |
| w/o IAB | 11.60 | 10.97 | 14.01 | 65.98 | 54.54 | 43.79 |
| Cross | 11.72 | 11.09 | 14.11 | **66.82** | 54.87 | 43.74 |
| IAB$^{*}$ | **11.86** | **11.13** | **14.21** | 66.11 | **54.92** | 43.90 |

**Table 4: Ablation of the number of queries. The number of queries $N_q$ is set within the range of $[5, 13]$.**

| $N_q$ | Relation Detection | | | Relation Tagging | | |
|---|---|---|---|---|---|---|
| | mAP | R@50 | R@100 | P@1 | P@5 | P@10 |
| 5 | 11.62 | 11.02 | 13.98 | 66.59 | 54.06 | 43.16 |
| 7 | 11.82 | 11.08 | 14.10 | 66.23 | **55.17** | 43.93 |
| 9$^{*}$ | **11.86** | **11.13** | **14.21** | 66.11 | 54.92 | 43.90 |
| 11 | 11.66 | 11.00 | 13.98 | 66.11 | 54.85 | 43.89 |
| 13 | 11.59 | 11.03 | 14.16 | **66.95** | 54.47 | 43.97 |

## 4.2 Ablation Studies

In this section, we conduct comprehensive ablation studies to demonstrate the effectiveness of the proposed Subject-Object Synergy Module. Additionally, we evaluate several critical parameters to affirm the robustness of our method.

**Subject-Object Synergy Module.** In Table 3, we present three variants to illustrate the effectiveness of the Subject-Object Synergy (SOS) module, including the removal of SOS and two different implementations of SOS (cross-attention vs. Interactive Attention Block). Without SOS, the results demonstrate a substantial drop (-0.32% mAP and -0.24% P@1) in detection and classification accuracy, indicating the importance of capturing the temporal feature patterns. Moreover, our IAB achieves superior perception of temporal and spatial representation within video clips, showing a notable advantage (+0.26 mAP) compared to the basic cross-attention.

**Table 5: Ablation of video length. Input videos are cropped to a unified length from a range of 256 to 1024.**

| Video Length | Relation Detection | | | Relation Tagging | | |
|---|---|---|---|---|---|---|
| | mAP | R@50 | R@100 | P@1 | P@5 | P@10 |
| 256 | 11.72 | 11.04 | 14.01 | **66.95** | **55.18** | 43.89 |
| 512* | **11.86** | **11.13** | **14.21** | 66.11 | 54.92 | 43.90 |
| 1024 | 11.71 | 11.10 | 14.05 | 65.50 | 54.73 | **44.17** |

**Table 6: Ablation of the number of the Subject-Object Synergy (SOS) module. Different numbers of the SOS modules ranging from 1 to 3 are stacked in the test.**

| No. of SOS | Relation Detection | | | Relation Tagging | | |
|---|---|---|---|---|---|---|
| | mAP | R@50 | R@100 | P@1 | P@5 | P@10 |
| 1 | 11.76 | **11.16** | 14.18 | 65.87 | 54.30 | 43.52 |
| 2* | **11.86** | 11.13 | **14.21** | **66.11** | 54.92 | **43.90** |
| 3 | 11.61 | 11.13 | 14.15 | 65.87 | **55.17** | 43.53 |

**Number of Queries.** The number of queries determines the capability of the video reasoning. A tight setting of $N_q$ hinders the modeling of diverse relationships, while an excessive query number results in redundant training complexity. Consequently, selecting an appropriate $N_q$ significantly affects its performance. Previous works [11] based on two-stage detection typically leverage a large query number (*e.g.*, $N_q = 100$) to simultaneously detect the relations of all the subject-object pairs. Distinct from that, our work independently estimates the relation for each pair, therefore requiring fewer queries, as demonstrated in Table 4. This can be extensively supported by a quantitative evaluation indicating that, on average, each subject-object pair in a single video clip in the VidOR training dataset is associated with 2.30 relations.

**Video length.** Table 5 shows the impact of the input length of the video clips. Empirically, we truncate/pad the videos to a uniform length of 512 for best performance.

**Number of SOS modules.** Table 6 illustrates the influence of the number of the Subject-Object Synergy layers. Accordingly, we set the number of layers as 2 in practice.

### 4.3 Qualitative Results

In Fig. 5, we present several visualization examples for comparison with BIG-C, VidVRD-II, and our VrdONE. The top part of Fig. 5 exhibits a sophisticated scene that features multiple humans and heavily occluded objects from VidOR dataset. Nonetheless, our VrdONE precisely captures the most of the relations. Specifically, our method can simultaneously consider spatial relations and action relations, *e.g.* "*adult-play(instr)-guitar*" and "*guitar-in front of-adult*", demonstrating that our method adequately considers spatiotemporal variance. In contrast, BIG [11] and VIDVRD II [30] are afflicted by the missed and wrong detections, especially in the case of the human and object interaction like "*adult-play(instr)-guitar*". In another easier case drawn from the VidVRD dataset, our VrdONE can produce diverse and confident detection results. It is worth mentioning that VrdONE accurately comprehends size and location relationships, affirming its advanced spatiotemporal understanding.

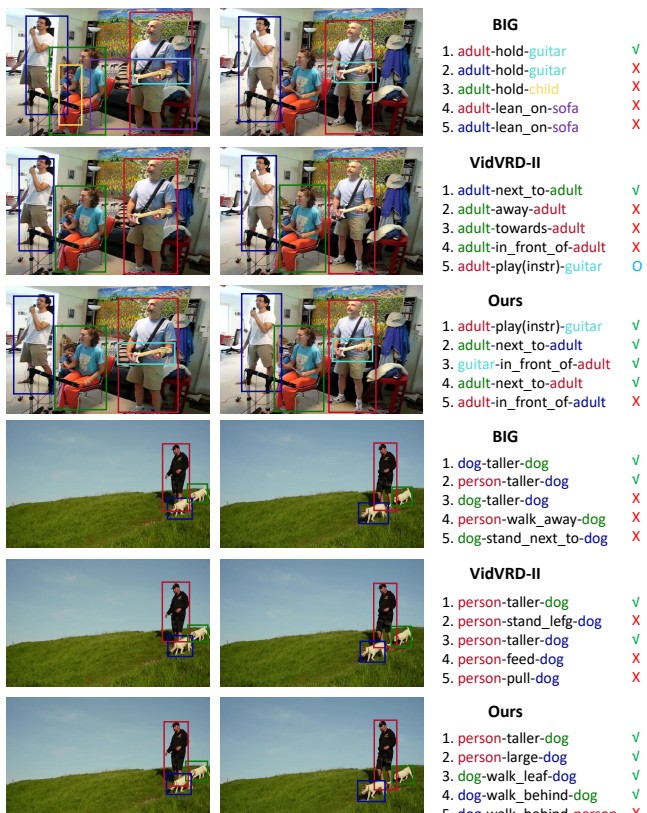

**Figure 5: Visualization of video relation detection and relation tagging results with open-source methods on VidOR dataset (top) and VidVRD dataset (bottom). The √, ×, and ◯ represent correct, false and missing detection respectively.**

Based on the results of the qualitative experiments above, we can fully demonstrate the superiority of our method and the effectiveness of spatiotemporal synergistic learning.

## 5 CONCLUSION

In this paper, we reframe the Video Visual Relation Detection challenge as a 1D instance segmentation problem and unveil VrdONE, a pioneering one-stage detection model designed to curtail redundant heuristic post-processing. By leveraging the dynamic interplay between subject-object pairs, VrdONE enhances video representation, improving both temporal classification and localization tasks. The novel Subject-Object Synergy (SOS) module within VrdONE adeptly captures both transient and lasting relations by synthesizing mutual features. Comprehensive quantitative and qualitative assessments affirm that VrdONE achieves unparalleled performance in its field.

**Limitations.** Despite VrdONE's advanced capabilities, it does exhibit certain constraints. Its effectiveness is partly dependent on the quality of the underlying pretrained video detection and tracking algorithms, as it utilizes processed tracklets for input. Additionally, VrdONE processes all possible subject-object pairs during inference without any preliminary filtering, potentially diminishing its overall efficiency.

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
