# OpenReview forum: "VrdONE: One-stage Video Visual Relation Detection"
_acmmm.org/ACMMM/2024/Conference — MM2024 Poster_

### Official Review · Reviewer_zmzP · 2024-05-24

**Rating:** 3
**Confidence:** 4

**Summary:**

This paper introduces VrdONE, a one-stage model for Video Visual Relation Detection. Unlike traditional methods that split category identification and temporal boundary detection, VrdONE integrates these processes by combining subject and object features into 1D instance segmentation. It also introduces the Subject-Object Synergy (SOS) Module, enhancing mutual perception before feature combination. VrdONE improves the performance of existing VidVRD models.

**Strengths:**

1. This paper proposed a novel approach that simultaneous category identification and binary mask generation for video relations.
2. This paper introduces VrdONE, which is the first one-stage framework for VidVRD.
3. The proposed model outperforms VidVRD on the ImageNet-VidVRD and VidOR datasets.

**Limitations:**

1. The writing of the paper is poor, and section 3 is not explained clearly.
2. The representation in the pipeline diagram appears unclear, please revise the diagram to align with the description provided in your methodology.
3. In the problem setting of section 3.1, it sets M possible relationships, but why is K used for the number of relationships in formula (1)?
4. BAS converts relational information into feature embeddings. What is its relationship with transient and persistent?
5. Formula (12) is not explained clearly.

**Suitability:**

3

---

### Official Review · Reviewer_XbuF · 2024-05-24

**Rating:** 4
**Confidence:** 3

**Summary:**

This paper introduces a novel perspective on video visual relationship detection (VidVRD) by conceptualizing temporal localization as a one-dimensional instance segmentation task. A new method named VrdONE is proposed, which integrates a bilateral spatiotemporal aggregation module and a one-stage relation detector. Experimental results on the VidVRD and VidOR datasets indicate that VrdONE achieves good performance.

**Strengths:**

1. This paper proposes addressing temporal localization in VidVRD as a novel one-dimensional instance segmentation task.
2. The proposed VrdONE method achieves good performance for VidVRD.
3. The paper is written in a detailed and clear manner.

**Limitations:**

1. The improvement brought by VrdONE is not highly significant.
2. The novelty of this paper primarily lies in the one-dimensional temporal instance segmentation; however, it lacks ablation studies or analyses to validate the advantages of this new paradigm.
3. For a one-stage method, it is recommended that the paper demonstrate its advantages in terms of inference time.
4. The presentation of Figure 5 is ambiguous and difficult to understand. It is recommended to use dividing lines or boxes to separate the top/bottom and detection/tagging areas.

**Suitability:**

3

---

### Official Review · Reviewer_T9Th · 2024-06-05

**Rating:** 4
**Confidence:** 2

**Summary:**

Summary: This paper introduces a framework for visual relation detection in videos. This framework is a one-stage method that does not require temporal proposal. The details of the models are well explained. The experiments were conducted on two datasets on both relation detection and relation tagging. The results show that the proposed framework achieves improved performance as to the previous models.

**Strengths:**

This paper introduces to address the relation detection method as a 1D segmentation problem. This enables the model to predict the relation temporal information in one feed forward propagation. The authors presented ablations of the model, showing the importance of each model component. The authors also analyzed the impact of different model parameters.

**Limitations:**

This paper claims that it addresses the challenge of complexity of the conventional methods which are a two-stage approach. The authors showed numerical results, however, it is unclear if the proposed framework can address the complexity/efficiency challenge. Additionally, from Table 1 it can be seen that the proposed framework shows similar performance as the Socal Fabric on the Relation Tagging task.

Many sentences used in this paper seem to be generated by AI technologies, many wordings are not natural languages and thus  are not easy to understand. The English usage of this paper needs significant improvement.
Many of notations used throughout the papers are not explained. Table 4 – Table 6 are the analysis of the model parameters, not the ablation of these parameters.

**Suitability:**

2

---

### Meta-Review · Area_Chair_U2wv · 2024-07-03

**Recommendation:** Accept (Poster)
**Confidence:** 5

**Metareview:**

This paper proposes addressing temporal localization in Video Visual Relation Detection (VidVRD) as a novel one-dimensional instance segmentation task, simplifying the complexity of the conventional two-stage approach. The proposed VrdONE method achieves good performance for VidVRD. The paper is well-written.

Most of the concerns previously raised by reviewers have been addressed in the rebuttal. Compared with state-of-the-art methods, the proposed method achieves the best performance with a moderate number of parameters and the least inference time. In addition to the existing ablation study, the authors have added an extra ablation study on 1D temporal instance segmentation in the rebuttal. The authors should also incorporate these changes into the paper upon acceptance.